# Torque Range, a New Parameter to Evaluate New and Used Instrument Safety

Dario Di Nardo, Marco Seracchiani, Alessandro Mazzoni, Andrea Del Giudice *,
Gianluca Gambarini and Luca Testarelli

Department of Oral and Maxillo Facial Sciences, "Sapienza" University of Rome, 00161 Rome, Italy;
dario.dinardo@uniroma1.it (D.D.N.); marco.seracchiani@uniroma1.it (M.S.);
alessandro.mazzoni@uniroma1.it (A.M.); gianluca.gambarini@uniroma1.it (G.G.);
luca.testarelli@uniroma1.it (L.T.)
* Correspondence: Andrea.DelGiudice@uniroma1.it or andreadelgiudice96@gmail.com

**Featured Application: Torque Range should be considered a crucial parameter for describing instrument safety and could ensure treatments in a more efficient way.**

**Abstract:** The aim of the present study is to evaluate operative torque, torque at failure and the difference between these two values—the "torque range"—of two different NiTi files. We also sought to evaluate and compare these parameters for new and used files. Forty S-One 20.06 and forty M-Two 20.06 were tested, divided in four equal groups (n = 10) for each brand. Ten instruments from each brand performed three root canal treatments each on standardized extracted single-rooted teeth. Afterwards, each group was subjected to the following two tests: operative torque and torsional resistance. Mean values for all the tested groups were calculated. The results for new instruments showed that S-One developed significantly higher operative torque, but higher torsional resistance. The results for used instruments showed that the S-One 20.06 developed less operative torque and higher mean torsional resistance value compared to the M-Two. Moreover, the percentage reduction of both values was significantly higher for M-Two. The results of the present study showed a safer torsional behavior of the S-One. These results could be related to the heat treatment and the manufacturing process.

**Keywords:** nickel–titanium rotary files; endodontic; torsional resistance; heat treatment

---

## 1. Introduction

During root canal treatments (RCT) instruments undergo flexural and torsional stresses that could eventually result in the instrument separation [1–3]. According to Sattapan et al. [4], torsional failure is the most common cause of intracanal failure. This fracture occurs when the tip or another apical part of the instrument binds into the canal while the upper part continues to rotate. More precisely, the separation occurs when torsional stress exceeds the plastic limit of the metal. To date in literature several studies investigate the parameters that could influence the torsional resistance. These factors are the cross section, the metal mass of the instrument, the heat treatments and the metallurgical properties of the nickel titanium rotary (NTR) instruments [5].

The most common method to investigate the torsional resistance of an instrument is a machine described by International Organization for Standardization (ISO) 3630-1 [6]. This method of evaluation was created for testing stainless-steel (SS) manual instruments. The above mentioned, method has a twofold problem: the torsional load is concentrated in a specific point of the instrument, while in clinical practice it is distributed along the whole operative portion of the NTR instruments. Furthermore, the

test is performed in static condition, not representative of intracanal working of the files. For these reasons, torque generated during instrumentation should be considered. Therefore, the concept of "operative torque" was introduced in recently published studies [7].

Operative torque can be defined as the quantity of torque developed during instrument progression towards the apex. It is a real time measurement of dynamic forces needed to perform the shaping of the canal. This value is influenced by time needed and technique used to reach working length. Indeed, each clinician has a different sensitivity that results in different clinical use of nickel titanium (NiTi) rotary instruments (amplitude of pecking, intensity of brushing etc.) [8]. Moreover, the morphology of the root canal system should be taken into account while considering operative torque since it is influenced by several anatomic characteristics such as canal trajectories and width, hardness of dentin, etc. [9].

The static torsional resistance test shows values of torque that should not represent the NiTi rotary instruments clinical behavior. Despite the operative torque clearly represents the clinical behavior of the files, it depends on the anatomy complexities of the canal and the technique used by the operator. Moreover, it does not show the maximum amount of torque reachable by a specific portion of the instrument. Hence, in order to overcome the problems of these two tests, the concept of torque range has been proposed by a previously published study [10].

This new parameter investigates the mean values of operative torque test in order to compare them to the values registered during the classic torsional resistance test. From this comparison a range can be identified, which should be used as parameter to define the instrument safety. Despite in the current literature several factors influencing the torsional resistance and the operative torque have been studied; no published articles have evaluated the influence of instrumentation on both the above-mentioned parameters.

Therefore, aim of the present study is to compare operative torque and torque at failure of two different NiTi files using both new and used instruments. Moreover, to evaluate the torque range before and after root canal shaping.

## 2. Materials and Methods

Two different instruments, S-One 20.06 (Fanta Dental CO., Ltd., Shanghai, China) and M-Two 20.06 (Sweden & Martina, Padova, Italy) were tested and compared in the present study. For each brand 40 instruments were randomly divided, using simple randomization method, in four equal groups (n = 10) and subjected to the following tests:

### 2.1. Operative Torque Tests

Forty extracted single-rooted human teeth were selected to perform two operative torque tests for each brand.

The single-rooted teeth were extracted for periodontal and orthodontic reasons. The surgical procedures were performed in accordance to Chiapasco's "Manual of Oral Surgery" [11]. The teeth were previously analyzed by cone beam computed tomography (CBCT) (KaVo, Biberach, Germany) in axial, coronal and sagittal sections. The CBCT analysis was performed to ensure that the extracted teeth presented the same anatomy and numbers of canal. All the extracted teeth presented one canal (Vertucci type I) with a curvature smaller than 30° according Schneider's criteria. Teeth presenting sign of resorption, immature apex, root fractures or calcification were discarded. The selected teeth were cleaned of any organic tissue, stored in saline solution and then autoclaved at 121 °C for 20 min.

Ten new (operative torque group A) rotary files for both S-One and M-Two performed an operative torque test on single-rooted teeth.

An endodontic access and a manual glide path with a #15 K-file to the working length was performed to all the extracted teeth by the same operator. Afterwards, irrigation with sodium hypochlorite at 5% (NaOCl) was performed with a 30-gauge syringe. According to the manufacturer recommendation, files were rotated clockwise at 300 rpm with 2-Ncm maximum torque using an

endodontic torque recording motor (KaVo, Biberach, Germany) [12]. The torque value recording was performed every 1/10 seconds.

Afterwards ten new instruments for both brands were selected and used to shape three single-rooted teeth each. All teeth used for this part of the study presented the same characteristics of the teeth used for the operative torque (i.e., curvature smaller than 30°, Vertucci Type I anatomic configuration).

After the shaping, a post instrumentation operative torque test was performed following the above described procedures. (operative torque group B).

Mean torque values (Ncm) were recorded for both group A and group B.

### 2.2. Torsional Resistance Test

Twenty instruments from each of the two groups were selected to perform this test.

Ten brand new instruments for both S-One and M-Two were immediately subjected to a torsional resistance test (torque at fracture group A).

This test was performed by the use of a torque recording endodontic motor (KaVo, Biberach, Germany). The device was already validated in a previous study to assess its accuracy and reliability.

A fixed block of 3 mm tip of the file was assessed with a mixed autopolymerizing resin (DuraLay; Reliance Dental Mfg. Co, Alsip, IL, USA). This system was blocked using a vise [13]. Each file was rotated clockwise at a speed of 300 rpm until fracture occurred. The torque limit was set at 5.5 Ncm, to ensure recording measurements ranging from 0.1 to 5.5 Ncm. Torque at fracture (TaF) values were collected.

Afterwards ten new instruments for both brands were selected and used to shape three single-rooted teeth each. All teeth used for this part of the study presented the same characteristics of the teeth used for the operative torque (i.e., curvature smaller than 30°, Vertucci Type I anatomic configuration).

Therefore, these ten instruments from each brand (group B) were selected to perform the torsional resistance test following the above described procedures.

### 2.3. Torque Range and Reduction of Torsional Resistance

Torque range is calculated subtracting the mean values of torque at failure and mean operative torque of each instrument.

Two torque ranges were analyzed. One was calculated by subtracting the mean values of TaF and operative torque from group A. The second was calculated by subtracting group B mean values of TaF and operative torque.

A percentage reduction of torsional resistance was calculated from mean values for TaF of group A and B.

### 2.4. Statistical analysis

All the collected data were statistically analyzed using the SPSS 17.0 software (SPSS Incorporated, Chicago, IL, USA). Mean and standard deviation were calculated.

T test for each of the tested group was performed with level of significance set at 5%.

## 3. Results

Table 1 shows mean values for both group A and B operative torque tests and the variation between the group in percentage. None of the recorded torque values exceeded the selected torque limit (Figures 1 and 2).

**Table 1.** Operative torque test (Ncm). *p* value was set a 0.05. Different capital superscript letters indicates significant difference among the groups (*p* < 0.05).

|  | Group A | Group B | *p* | Variation between the Group in Percentage |
|---|---|---|---|---|
| S-One | 0.550 ± 0.010 [aB] | 0.470 ± 0.010 [aD] | 0.105 | −14.54% |
| M-Two | 0.345 ± 0.060 [BC] | 0.670 ± 0.020 [CD] | 0.022 | 194.20% |
| *p* | 0.043 | 0.012 | | |

Capital superscript letters indicate a significance difference. Lower case superscript letters indicate no significant difference.

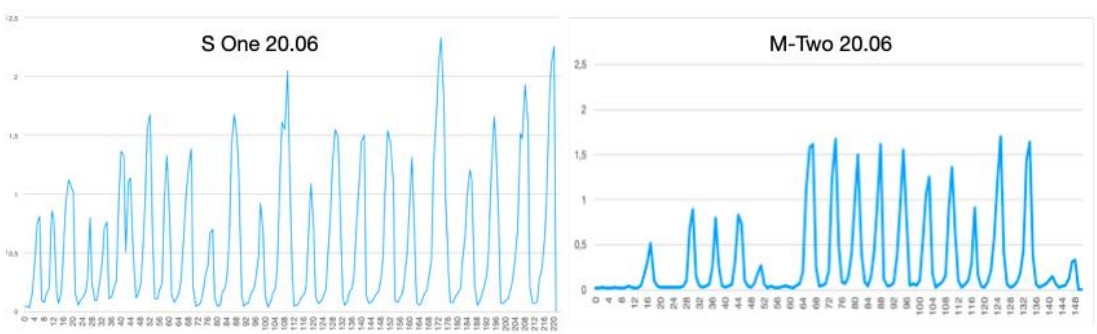

**Figure 1.** Operative torque for group A.

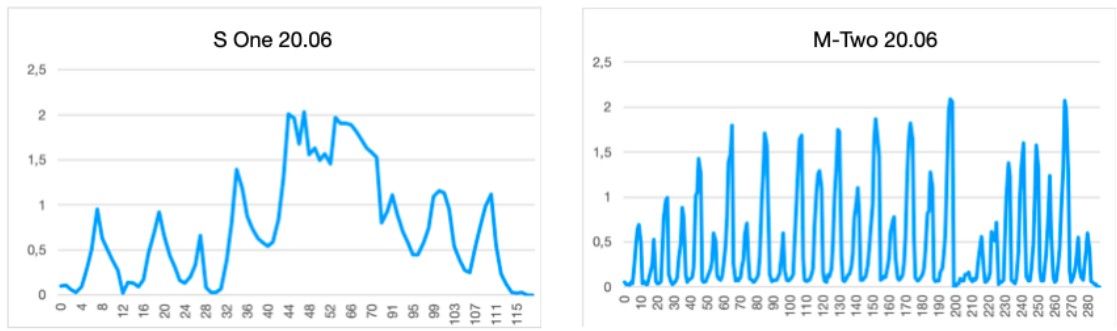

**Figure 2.** Operative torque for group B.

Compared to M-Two, S-One group A developed significantly higher mean torque values to reach working length.

After the ex-vivo instrumentation the M-Two generated significantly more operative torque compared to the S-One (*p* < 0.05).

In terms of torsional static resistance (torque at failure) the two files demonstrated significant statistical differences (*p* < 0.05), for both group A and B, as shown in Table 2. The amount of reduction in terms in static torsional resistance is showed also in Table 2.

Table 3 indicates the range between the mean values of torque at failure and operative torque of the instruments for both groups. Regarding group A, the M-Two has a safer range compared to the S-One. Instead, comparing the instruments in group B, the M-Two range becomes even negative.

**Table 2.** Mean values for torque at Failure (TaF). *p* value was set at 0.05. Different capital superscript letters indicates significant difference among the groups ($p < 0.05$).

|  | Group A | Group B | *p* | Amount of Reduction in Percentage |
|---|---|---|---|---|
| S-One | 0.760 ± 0.010 [AB] | 0.550 ± 0.020 [AD] | 0.011 | 27.6% |
| M-Two | 0.645 ± 0.010 [BC] | 0.425 ± 0.010 [CD] | 0.001 | 34.1% |
| *p* | 0.009 | 0.026 |  |  |

Capital superscript letters indicate a significance difference.

**Table 3.** Values of torque range.

|  | S-One | M-Two |
|---|---|---|
| Group A | 0.210 | 0.300 |
| Group B | 0.08 | −0.245 |

## 4. Discussion

In present literature, torsional resistance is the most used test to evaluate the metallurgical features of the alloy in a static rotational resistance [14]. Nonetheless, this type of test do not take into account the multitude of stresses, since NiTi rotary instruments are used dynamically [15]. In order to evaluate the instrument during motion, the operative torque could be a more valuable test. This dynamic measurement should be determined by time needed and technique used to reach the working length [7].

To ensure a precise evaluation of the instrument metallurgical features in terms of static torsional resistance and dynamic behavior, the concept of torque range was chosen in the present article.

Torque range shows the correlation between torsional stresses during instrumentation and instruments' resistance to them. It has a clinical relevance due to the fact that considers both the role of torsional static resistance and dynamic operative torque.

M-Two rotary files (Sweden & Martina, Padova, Italy) are an endodontic file manufactured with traditional nickel–titanium alloy without any heat treatment. The instruments present an "S shaped" cross sectional design with two active cutting blades along instrument surface. These rotary instruments are thought to be used in a sequence [16,17]. The pitch is variable and increases from the tip to the handle. In order to carry out the study the M-Two 20 tip and 0.06 taper was selected.

The S-One (Fanta Dental Co., Ltd., Shanghai, China) is a new NiTi rotary recently introduced in the market. This instrument has a similar S-Shaped cross-sectional design of the M-Two. Moreover, the instrument present a new heat treatment recently produced by the manufacturer the AF-H wire [18]. This, according to manufactured internal studies, ensure a more bendable file and more flexural and torsional resistance compared to other instruments.

The results describe the behavior of the two rotary instruments regarding the three parameters investigated.

For new files (group A), S-One showed significant higher torsional resistance compared to M-Two.

This result is in accordance with part of the current literature, indeed the role of thermal treatments on torsional resistance is not completely clarified [5,13]. Many studies declared that the presence of heat-treated files do not significant affect or even enhance, the torsional resistance of an instruments [5,19].

No published studies investigated the torsional resistance of an endodontic file before and after intracanal instrumentation. Therefore, it is pretty unknown the influence of instrumentation on torsional behavior. In this study, after the three single-rooted instrumentation (group B), torsional resistance of both M-Two and S-One showed a significant decrease value.

This result could be explained by the fact that repetitive usage of an instrument not only modify the external surface of the instrument, but also the internal surface. Therefore, the peculiar torsional

pattern of fracture starts from the center of the instrument not reaching the external surface. Indeed, despite also the external cutting surface of the instruments could present microscopic deformation after multiple intracanal instrumentation, these can be detected using a high microscope. Instead, the internal deformation caused by torsional stresses, lead to internal crack formations that cannot be detected. The presence of these microscopic deformations could reduce the torsional resistance of a NTR files.

Moreover, the AF-H wire, not only could allow the instrument to better withstand to torsional stress but could also explain the percentage of torsional resistance reduction (27.6%), if compared to the M-Two (34.10%).

The operative torque of brand-new files, shows a more efficient (i.e., less operative torque) dynamic shaping of the M-Two. Despite the multiple factors that influence the operative torque are not completely understood, this result can be explained by the greater cutting efficiency of the instrument.

The results of operative torque test for used instruments (group B) showed that the S-One 20.06 (Table 2) developed less operative torque (0.47 ± 0.01 Ncm) compared to the M-Two. This result is due to the difference in the effect of instrumentation on the two different files. Indeed, while used M-Two increased in a significant way the values of operative torque compared to news (+194.2%), the shaping seems to less affect the performance of S-One. No statistically relevant difference was found between operative torque for new and used S-One.

The increase of operative torque could be explained by the reduction of cutting ability of an NTR instruments after repetitive usage. In current literature many studies described the role of blunted blade during intracanal instrumentation [20]. The lack of sharpness increases operative time during RCT and reduce the cutting efficiency of an instrument. Moreover, the instrument would necessarily develop higher values of torque to progress inside the canal, cut dentin and remove debris.

The two parameters alone, despite describing the behavior of the NTR instrument in both static and dynamic condition, do not relate the in vivo shaping and the in vitro resistance conditions.

Therefore, torque range was introduced as a parameter to understand the performance of an instrument regarding torsional stresses.

It was developed to relate the mean values of operative torque test to the values registered during the classic torsional resistance test [6,21]. As a result, an instrument that exhibits a wide range is safer than an instrument that exhibits a shorter or even negative range (Figure 3).

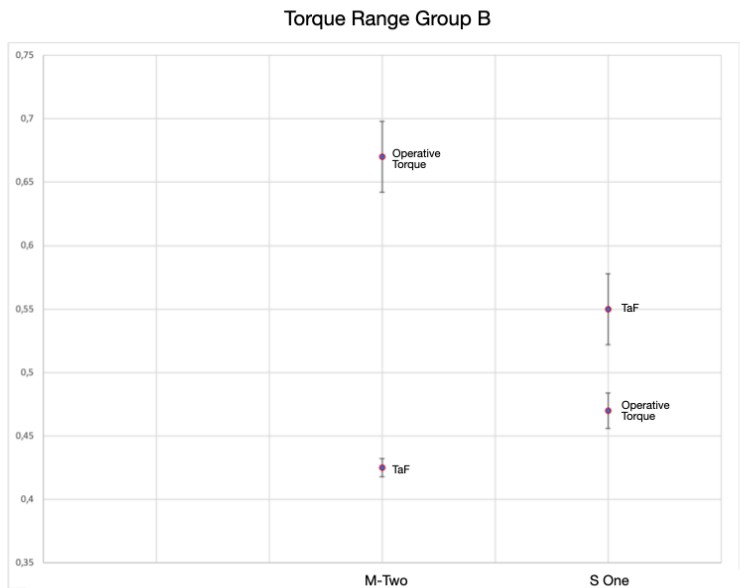

**Figure 3.** Torque range evaluation for M-Two 20.06 and S-One 20.06.

The S-One exhibited a wider range compared to the M-Two. Knowing the maximum deformation limit of the alloy in a determined portion of the instrument (torsional resistance) and the working condition, i.e., amplitude of pecking and apical pressure, during root canal instrumentation (operative torque), it could be possible to suggest a safety interval in which the instrument should act [22].

## 5. Conclusions

The present article investigated the role of the torque range over the torsional resistance of an instrument, both in a static and dynamic way. This parameter was also used to evaluate instrument reliability in terms of percentage of reduction to torsional resistance. Despite the limitation of the study performed on ex vivo samples, the torque range should be considered a crucial parameter for describing instrument safety: wider the range, safer the instrument. The S-One 20.06 exhibited a wider range than the M-Two 20.06, showing a better performance both before and after multiple usage. Torque range should represent a possible method of clinical evaluation of safety during RCT.

**Author Contributions:** Conceptualization, L.T.; methodology, M.S.; investigation, A.D.G.; resources, A.M.; data curation, A.D.G. and A.M.; writing—original draft preparation, D.D.N.; writing—review and editing, G.G.; visualization, M.S.; All authors have read and agreed to the published version of the manuscript.

**Funding:** This research received no external funding.

**Conflicts of Interest:** The authors declare no conflict of interest.

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
