# Peer review of "Torque Range, a New Parameter to Evaluate New and Used Instrument Safety"

_applsci, doi:10.3390/app10103418_

Round 1

Reviewer 1 Report

The paper titled “Torque range, a new parameter to evaluate new and used instrument safety” is an interesting work that compares operative torque and torque at failure of two different NiTi files. Nevertheless, this paper has some minor errors.

Minor errors:

1). In abstract. It must be rewritten to make it more understandable to the reader. In addition, references to tables must be deleted.

2). In references. The authors must read the instructions for authors. The number of the journal in parenthesis should be deleted.

Author Response

Response to Reviewer 1 Comments

Point 1:In abstract. It must be rewritten to make it more understandable to the reader. In addition, references to tables must be deleted.

Response 1: Abstract has been completely rewritten. Therefore, references to “table” in abstract have been deleted.

Point 2:In references. The authors must read the instructions for authors. The number of the journal in parenthesis should be deleted.

Response 2: Reference style has been changed as recommended.

Reviewer 2 Report

The present study is well presented, however, some minor points should be addressed as follows: 

  1. Line 14. This in order to show....this sentence must be improved
  2. Line 31. RCT is an abbreviation for root canal treatmentas stated above; therefore saying during a RCT is wrong
    undergo to? I would leave to 
  3. Line 38. First time when you are mentioning Ni-Ti, the full name should be inserted (nickel-titanium)
  4. Line 41. T The
  5. Line 52. Each shouldn't be written with a capital letter
  6. Why are expressions such as operative torque and torque range written with capital letters? 
  7. Introduction needs to be improved. There are to many paragraphs and text does not flow correctly 
  8. Line 81. Write human teeth so that we do not read further to be sure what kind of teeth were used
  9. Where were the teeth kept?
  10. How did you prepare endodontic access? Who did perform it? Always the same person? Did you irrigate?
  11. How did you randomize the samples? 
  12. Line 89. 121°C
  13. Figure legend should also include a statement of the results to ensure clarity. 
  14. Line 133. Mean Values (Why capital letters?)
  15. Table 1 - not aligned properly
  16. The decimal separator should always be the same (use the dot)
  17. Line 154- motion, the 
  18. Line 181. ...in accordance with part of the literature - mention those papers, please
  19. Lines 182, 190, 194, 212. ....double space
  20. Tables - it would be better to use this symbol ±
  21. Reference style- you do not need numbers in brackets 

Author Response

Response to Reviewer 2 Comments

Point 1: Line 14. This in order to show...this sentence must be improved.

Response 1: Abstract has been rewritten.

Point 2: Line 31. RCT is an abbreviation for root canal treatmentas stated above; therefore saying during a RCT is wrong.

undergo to? I would leave to 

Response 2: Abbreviation has been corrected, Line 29; “to” has been deleted Line 29

Point 3: Line 38. First time when you are mentioning Ni-Ti, the full name should be inserted (nickel-titanium)

Response 3: Done Line 36

Point 4: Line 41. T The

Response 4: Done

Point 5: Line 52. Each shouldn't be written with a capital letter

Response 5: Done

Point 6: Why are expressions such as operative torque and torque range written with capital letters?

Response 6: Capital letters of expression such as Group, Operative Torque, Torque Range, Torsional Resistance and Mean Values were changed.

Point 7: Introduction needs to be improved. There are to many paragraphs and text does not flow correctly 

Response 7: The whole introduction has been modified to keep it more fluent. Paragraphs have been reduced from 13 to 6.

Point 8: Line 81. Write human teeth so that we do not read further to be sure what kind of teeth were used

Response 8: Done

Point 9: Where were the teeth kept?

Response 9: According to Nawrocka A et al. Teeth were stored in saline solution and autclavated at 121°C before performing the ex vivo tests. “Nawrocka A, Łukomska-Szymańska M. Extracted human teeth and their utility in dental research. Recommendations on proper preservation: A literature review.Dent Med Probl. 2019 Apr-Jun;56(2):185-190. We added this information in the manuscript as follow “The selected teeth were cleaned of any organic tissue, stored in saline solution and then autoclaved at 121°C for 20 min” Line 147

Point 10: How did you prepare endodontic access? Who did perform it? Always the same person? Did you irrigate?

Response 10: We added these informations in manuscript as follow: “…performed to all the extracted teeth by the same operator. Afterwards, irrigation with Sodium Hypoclorite at 5% (NaOCl) was performed with a 30 gauge syringe.” Line 152

Point 11: How did you randomize the samples?

Response 11:  We added this information in the manuscript as follow “…40 instruments were randomly divided, using simple randomization method, in four equal…” Line 135

Point 12: Line 89. 121°C

Response 12: Done

Point 13: Figure legend should also include a statement of the results to ensure clarity.

Response 13: Done

Point 14: Line 133. Mean Values (Why capital letters?)

Response 14: Done

Point 15: Table 1 - not aligned properly

Response 15: Done

Point 16:The decimal separator should always be the same (use the dot)

Response 16: Done

Point 17: Line 154- motion, the 

Response 17: Changed

Point 18: Line 181. ...in accordance with part of the literature - mention those papers, please

Response 18: Done

Point 19: Lines 182, 190, 194, 212. ....double space

Response 19: Changed

Point 20: Tables - it would be better to use this symbol ±

Response 20: Done

Point 21: Reference style- you do not need numbers in brackets

Response 21: Done 

Reviewer 3 Report

The topic of this article entitled “Torque Range, a New Parameter to Evaluate New and Used Instrument Safety.” is aimed to compare operative torque and torque at failure of two different NiTi files. I ask authors to add some key concepts. - The authors have reported test on extracted teeth, however, nothing has been reported on clinical procedure to manage patients, please give some clarifications. The authors must describe other type of tests to evaluate stress on different biomaterials, such as the three-points bending test, so to create some background to their work (for example, discuss: Marrelli, M.; Maletta, C.; Inchingolo, F.; Alfano, M.; Tatullo, M. Three-point bending tests of zirconia core/veneer ceramics for dental restorations. Int J Dent 2013, 2013, 831976.). - Authors should also give their point of view on nanotechnologies and nanomaterials on medical procedures and their clinical applications (for example, discuss: Barry M, Pearce H, Cross L, Tatullo, M., Gaharwar AK. Advances in Nanotechnology for the Treatment of Osteoporosis. Curr. Osteoporos Rep 2016, 14, 87–94.) A graphical abstract would be helpful to better explain your work. A slight improvement of comparation among different similar instruments would help readers to understand the novelty of your interesting work. Conclusions should report limitations of this study and potential applications of this research to future clinical applications.

Author Response

Response to Reviewer 3 Comments

The topic of this article entitled “Torque Range, a New Parameter to Evaluate New and Used Instrument Safety.” is aimed to compare operative torque and torque at failure of two different NiTi files. I ask authors to add some key concepts.

Point 1: The authors have reported test on extracted teeth, however, nothing has been reported on clinical procedure to manage patients, please give some clarifications.

Response 1: Clinical procedures to manage patients have been indicated: ”The surgical procedures were performed in accordance to Chiapasco’s “Manual of Oral Surgery” Line 148

Point 2: The authors must describe other type of tests to evaluate stress on different biomaterials, such as the three-points bending test, so to create some background to their work (for example, discuss: Marrelli, M.; Maletta, C.; Inchingolo, F.; Alfano, M.; Tatullo, M. Three-point bending tests of zirconia core/veneer ceramics for dental restorations. Int J Dent 2013, 2013, 831976.). - Authors should also give their point of view on nanotechnologies and nanomaterials on medical procedures and their clinical applications (for example, discuss: Barry M, Pearce H, Cross L, Tatullo, M., Gaharwar AK. Advances in Nanotechnology for the Treatment of Osteoporosis. Curr. Osteoporos Rep 2016, 14, 87–94.) A graphical abstract would be helpful to better explain your work.

Response 2: We apologize but in our humble opinion it is not worth discussing about zirconia core/veneer ceramics and nanotechnologies. Moreover different tests available for nickel titanium have been already cited in the manuscript. We are not able for this type of study, to produce a graphical abstract. Moreover, no published studies present this kind of images.

Point 3: A slight improvement of comparation among different similar instruments would help readers to understand the novelty of your interesting work.

Response 3: Comparison of the instruments has been written as follow: “M-Two rotary files (Sweden & Martina, Padova, Italy) are an endodontic file manufactured with traditional Nickel Titanium alloy without any heat treatment. The instruments present an “S shaped” cross sectional design with two active cutting blades along instrument surface. These rotary instruments are thought to be used in a sequence [16, 17]. The pitch is variable and increases from the tip to the handle. In order to carry out the study the M-Two 20 tip and 0.06 taper was selected. 

The S One (Fanta Dental Co., Ltd, Shanghai, China) is a new NiTi rotary recently introduced in the market. This instrument has a similar S-Shaped cross sectional design of the M-Two. Moreover, the instrument present a new heat treatment recently produced by the manufacturer the AF-H wire [18]. This, according to manufactured internal studies, ensure a more bendable file and more flexural and torsional resistance compared to other instruments.”

Point 4: Conclusions should report limitations of this study and potential applications of this research to future clinical applications.

Response 4: Limitation of the studies and further clinical application have been indicated in the manuscript as follow:” Despite the limitation of the study performed on ex vivo samples, the torque range should be considered a crucial parameter for describing instrument safety: wider the range, safer the instrument. The S-One 20.06 exhibited a wider range than the M-Two 20.06, showing a better performance both before and after multiple usage. Torque range should represent a possible method of clinical evaluation of safety during RCT.”

Round 2

Reviewer 3 Report

This very interesting and highly impacting article can be published in Applied Science.